# Single-cell RNA sequencing of human nail unit defines *RSPO4* onychofibroblasts and *SPINK6* nail epithelium

Hyun Je Kim [1,2,11], Joon Ho Shim [1,3,11], Ji-Hye Park [1], Hyun Tae Shin [4], Jong Sup Shim[5], Kee-Taek Jang [6], Woong-Yang Park [3], Kyung-Hoon Lee [7], Eun Ji Kwon [8], Hyung-Suk Jang[1], Hanseul Yang [9], Jong Hee Lee[1,10], Jun-Mo Yang[1] & Dongyoun Lee [1✉]

Research on human nail tissue has been limited by the restricted access to fresh specimen. Here, we studied transcriptome profiles of human nail units using polydactyly specimens. Single-cell RNAseq with 11,541 cells from 4 extra digits revealed nail-specific mesenchymal and epithelial cell populations, characterized by *RSPO4* (major gene in congenital anonychia) and *SPINK6*, respectively. In situ RNA hybridization demonstrated the localization of *RSPO4, MSX1 and WIF1* in onychofibroblasts suggesting the activation of WNT signaling. *BMP-5* was also expressed in onychofibroblasts implicating the contribution of BMP signaling. *SPINK6* expression distinguished the nail-specific keratinocytes from epidermal keratinocytes. *RSPO4*[+] onychofibroblasts were distributed at close proximity with *LGR6*[+] nail matrix, leading to WNT/β-catenin activation. In addition, we demonstrated *RSPO4* was overexpressed in the fibroblasts of onychomatricoma and *LGR6* was highly expressed at the basal layer of the overlying epithelial component, suggesting that onychofibroblasts may play an important role in the pathogenesis of onychomatricoma.

[1] Department of Dermatology, Samsung Medical Center, Sungkyunkwan University School of Medicine, Seoul, Republic of Korea. [2] Institute of Endemic Diseases, College of Medicine, Seoul National University, Seoul, Republic of Korea. [3] Samsung Genomic Institute, Samsung Medical Center, Sungkyunkwan University School of Medicine, Seoul, Republic of Korea. [4] Department of Dermatology, Inha University School of Medicine, Incheon, Republic of Korea. [5] Department of Orthopedic Surgery, Samsung Medical Center, Sungkyunkwan University School of Medicine, Seoul, Republic of Korea. [6] Department of Pathology, Samsung Medical Center, Sungkyunkwan University School of Medicine, Seoul, Republic of Korea. [7] Department of Anatomy, Sungkyunkwan University School of Medicine, Suwon, Republic of Korea. [8] Morristown Pathology Associates, Morristown, NJ, USA. [9] Department of Biological Science, Korea Advanced Institute of Science and Technology, Daejeon, Republic of Korea. [10] Department of Medical Device Management & Research, Samsung Advanced Institute for Health Sciences & Technology, Sungkyunkwan University, Seoul, Republic of Korea. [11]These authors contributed equally: Hyun Je Kim, Joon Ho Shim. ✉email: dylee@skku.edu

The nail unit is one of the major skin appendages. It is comprised of epithelial and subjacent mesenchymal elements[1]. The epithelial components include the nail matrix, nail bed, and nail plate. The nail plate is composed by maturation and keratinization of the cells from the nail matrix and attached to the nail bed, which may also contribute to the nail formation[2].

The nail and hair units are both hard keratin generating organs that share many commonalities with respect to their embryogenesis, anatomy, and co-involvement in various diseases[3,4]. The hair unit has been more extensively studied due to easier access to research materials compared to the nail unit, ant it produces hair via coordinated interactions between specialized mesenchyme cells known as dermal papilla and epithelial stem cells[5]. We have been studying the nail unit to identify analogous mesenchymal–epithelial interactions, specifically investigating the specialized mesenchymal portion of the nail unit which we propose are responsible for communicating with nail stem cells.

In line with this endeavor, we identified a CD10-positive mesenchymal cell population underneath the nail matrix and nail bed and proposed to name these specialized mesenchymal cells as onychofibroblasts[6]. We also documented the presence of a specialized onychofibroblast-containing mesenchyme that demonstrates histologic and immunohistologic features that are distinguishable from another part of the nail unit[7]. Based on these findings, we hypothesized that onychofibroblasts may play an important role in nail formation and growth through epithelial–mesenchymal interactions.

In an attempt to obtain a holistic view of the potential functions of onychofibroblasts, we profiled transcriptomes of nail units from four fresh human polydactyly specimens using single-cell RNAseq (scRNAseq). We also investigated the gene expressions in nail-specific epithelial and mesenchymal cells.

## Results

**Single-cell RNAseq of human nail units**. We strategically enriched single cells from the nail matrix and portions of the nail unit containing onychofibroblasts by removing volar skin, distal phalangeal bone and nail plate (Fig. 1a) since single cells from the non-nail unit could weaken the nail-specific signatures in the dataset. A total of 11,541 cells with a median of 2354 genes per cell from 4 polydactyly samples (515–4023 cells/sample; Supplementary Fig. 1a, b; Supplementary Table 1) were profiled. None of the polydactylous patients reported other phenotypes or genetic mutations. Uniform manifold approximation and projection (UMAP) plot reflected 18 conserved clusters (Fig. 1b, c; Supplementary Fig. 1c). Canonical markers clearly demarcated each cell cluster (Fig. 1d): *KRT*1 or *KRT5* for keratinocytes; *COL1A1* for fibroblasts; *RSG5* for myofibroblast-like cells; *VWF* for endothelial cells; *LYVE-1* for lymphatic endothelial cells; *PTPRC* for lymphocytes; *AIF1* for macrophage/dendritic cells; *TPSB2* for mast cells; *SCGB1B2P* for eccrine gland cells; *DEFB1* for eccrine duct cells; *COL9A2* for chondrocytes; *NRXN1* for neural cells, and *PMEL* for melanocytes[8–12]. The full list of differently expressed genes associated with each cell-type cluster is presented in Supplementary Data 1 and Supplementary Fig. 1d. Three to five supervised gene sets representing each cluster deduced from literature search are summarized in Supplementary Fig. 2.

**Subclustering of fibroblasts identifies the nail-specific cell population (onychofibroblasts)**. To further analyze the heterogeneity of nail unit fibroblasts, we extracted the fibroblast cluster and performed sub clustering analysis (Fig. 2a). Based on our previous observation of positive *MME* (CD10) expression on the

onychofibroblasts within the onychodermis[7,13], we investigated the expression of *MME* amongst the different fibroblasts clusters. Violin and feature plot demonstrated a distinct cluster of fibroblasts (Fibroblast 4 cluster) demonstrating a high *MME* expression level. Hence, we defined this cluster as onychofibroblasts. Next, we sought to investigate the expression of *RSPO4*, a member of the R-spondin family of a secreted protein involved in WNT signaling as it is implicated in congenital anonychia[14]. We observed that the onychofibroblast cluster also demonstrated a high expression of *RSPO4* (Fig. 2b). We also found additional genes that were strongly and specifically expressed in the onychofibroblast cluster including *MSX1*, *TWIST1*, *CRABP1*, *WIF1*, and *BMP5* (Fig. 2c). Gene Ontology enrichment analyses showed that genes involved in skeletal system development, ossification, and extracellular structure organization are significantly enriched in the onychofibrobalst (fibroblast 4 clusters) compared to fibroblast 1–3 clusters (Fig. 2d).

**In situ RNA hybridization confirms gene expression patterns of onychofibroblasts revealed by scRNAseq**. To confirm the expression of *RSPO4* in the nail unit, we performed in situ RNA hybridization (ISH) on polydactyly samples. ISH demonstrated *RSPO4* mRNA expression restricted to mesenchymal cells below the nail matrix and nail bed with no expression in dermal fibroblasts elsewhere around the nail unit (Fig. 2e). Adult nail biopsy also showed onychofibroblasts expressing *RSPO4* mRNA within the onychodermis (Fig. 2f). Other markers, including *BMP5*, *MSX1*, and *WIF1*, were also expressed by the onychofibroblasts (Fig. 2g). The distribution of these onychofibroblasts ranged from the dermis subjacent to the nail matrix keratinocytes and to the underlying deeper dermis. Hematoxylin and eosin (H&E) staining and CD10 immunohistochemical stain suggest the existence of two different cell types: superficially located wavy-shaped onychofibroblasts and deep dermal located round to oval-shaped onychofibrobalsts (Supplementary Fig. 3a–e). This heterogeneity of onychofibroblasts was also supported by the differential expression of onychofibroblast-specific genes (*MSX1*, *BMP5*, *WIF1*, *TWIST1*, *CRABP1*, and S*FRP2*, Supplementary Fig. 4a–c).

**Subclustering of keratinocytes identifies the nail-specific cell populations**. Next, we focused on the epithelial cell populations to define nail-specific keratinocytes responsible for nail formation. A total of six epithelial subpopulations were identified from polydactyly specimens (Fig. 3a). Basal keratinocytes were defined by the expression of *KRT5* and *KRT14*, suprabasal keratinocytes by *KRT1* and *KRT10* expression, proliferating keratinocytes by *MKI67* expression, and cornified cells by *SPRR1B* expression[15].

We further identified two keratinocyte subpopulations in the polydactyly specimens demonstrating high expression of *SPINK6* (Supplementary Fig. 5a) and *WNT6* (Fig. 3b). To address whether these keratinocyte subpopulations are nail-specific, we obtained transcriptome data for 2758 single keratinocytes from a published cohort of skin samples and integrated it with a polydactyly dataset (Fig. 3b; Supplementary Fig. 6a, c; Supplementary Table 2)[11]. As an expression of these latter two genes was rarely observed in the keratinocytes derived from the skin samples, we thought these keratinocyte subpopulations are nail-specific (Fig. 3b; Supplementary Fig. 5a). *SPINK6*+ keratinocytes mainly expressed *KRT16* suggesting that they might be derived primarily from nail bed[16]. These major cell types of the nail unit including onychofibroblasts were also represented in the individual analysis (Supplementary Fig. 7).

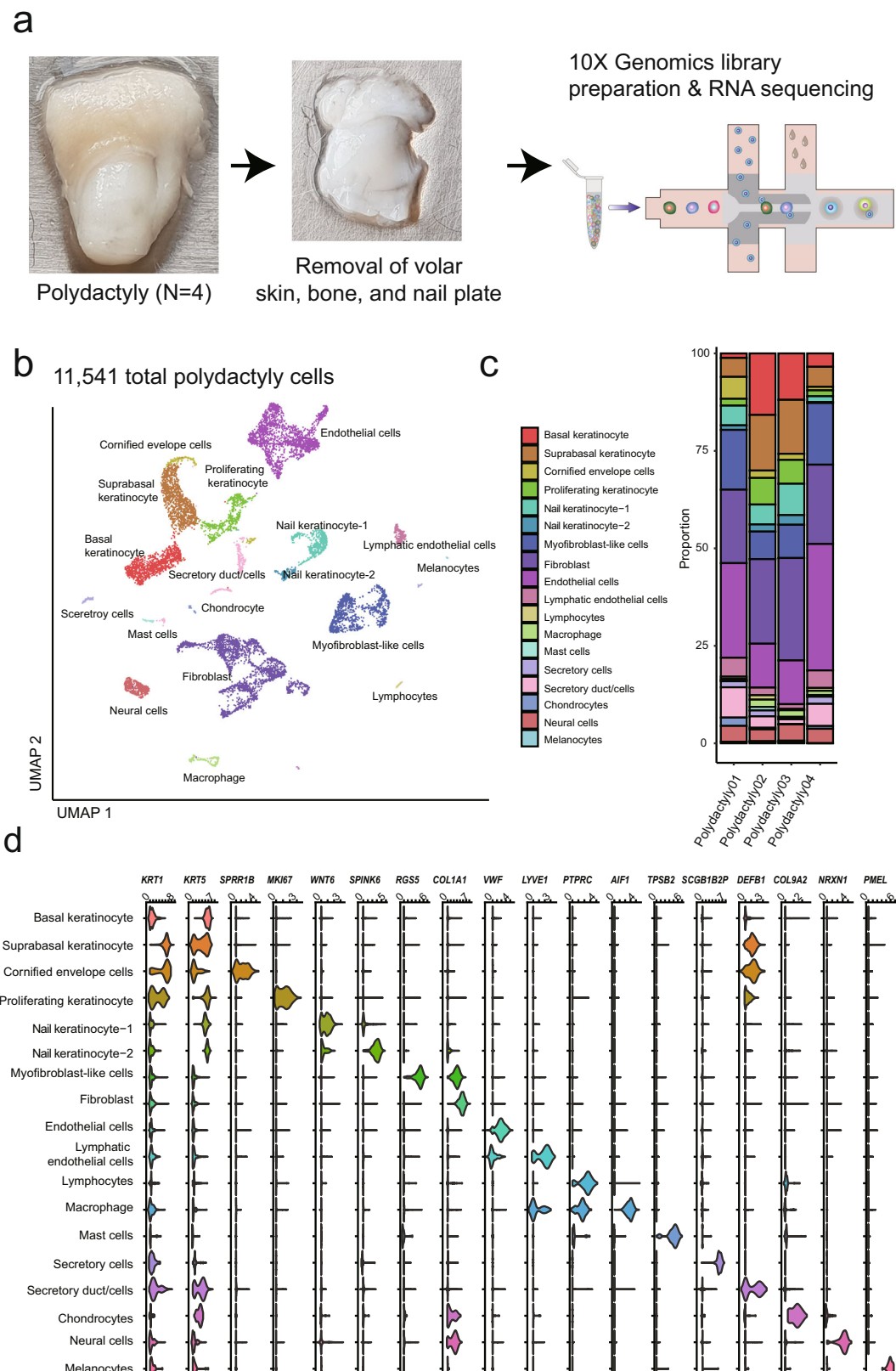

**Fig. 1 scRNAseq of human nail unit reveals conserved cell populations. a** An overview. Four extra-digits were dissociated into the single-cell suspension ($n = 11,541$ cells). **b** A UMAP plot demonstrated 18 clusters. **c** Relative proportions of each cell-type color-coded by cell-type and annotated for clusters as in Fig. 1b. **d** Cluster annotation. The violin plot represents an expression of canonical markers of each cluster.

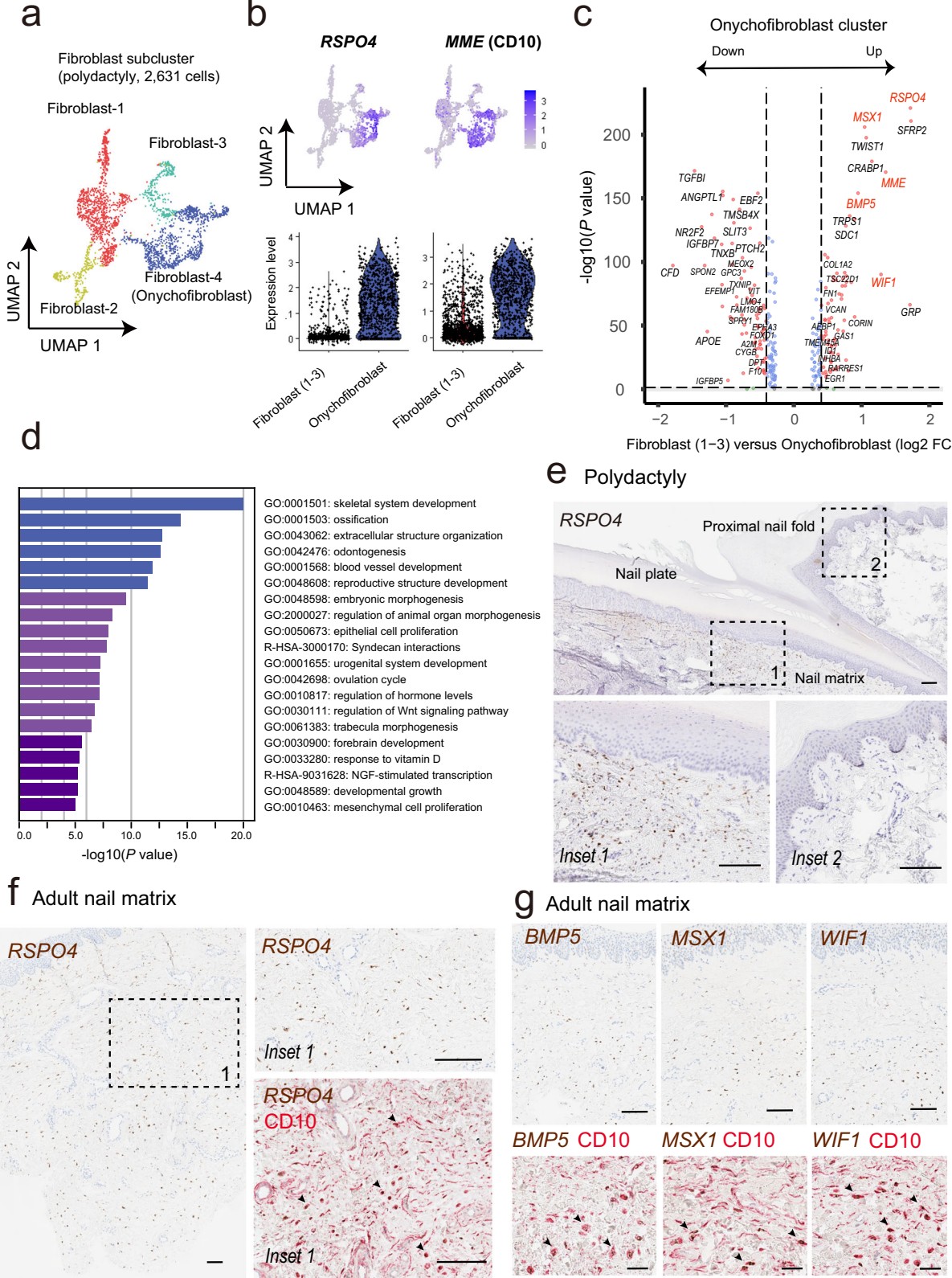

**Immunostaining and ISH confirm gene expression patterns of nail-specific keratinocytes identified by scRNAseq.** To validate the localization of *SPINK6* positive nail keratinocytes, immunohistochemistry (IHC) was done on the sections of normal nail units. IHC showed *SPINK6* expression in the nail epithelium while its expression was not found in the non-nail unit epidermis (Fig. 3c). Considering the *SPINK* family's roles as a protease

inhibitor in keratinocyte differentiation, *SPINK6* might have a role in maintaining epithelial barrier in the nail unit[17]. Since *SPINK6* was reported to be expressed on normal human skin[17], we conducted Microarray and qPCR analyses to investigate whether nail epithelium expresses a much higher level of *SPINK6* compared to normal skin. As scRNAseq data suggested, we confirmed a specifically higher expression of *SPINK6* in nail

**Fig. 2 Subclustering of fibroblasts derived from polydactyly identifies the nail-specific cell population (onychofibroblasts). a** A UMAP of fibroblasts reveals four distinct clusters. **b** *RSPO4* and *MME* (CD10) expressions on onychofibroblasts. **c** Volcano plot represents the distinctively expressed gene sets on onychofibroblasts. **d** Gene Ontology enrichment analysis with differentially expressed genes in onychofibroblasts. **e** ISH of *RSPO4* on polydactyly tissue. *RSPO4* expression primarily observed beneath the nail matrix and nail bed. Scale bar = 100 μm. **f** ISH of *RSPO4* on adult nail matrix tissue. Intranuclear *RSPO4* transcripts were observed in the mesenchyme beneath the nail matrix epithelium. Co-label IHC for CD10 revealed co-expression of *RSPO4* and *MME* in onychofibroblasts (lower right). Scale bar = 100 μm. **g** ISH of *BMP5*, *MSX1*, and *WIF1* on adult nail matrix tissue. Scale bar = 100 μm. Co-label IHC for CD10 (lower image; scale bar = 20 μm). Additional ISH with higher magnifications is presented in Supplementary Fig. 3d. The list of ISH or IHC markers is provided on the upper left corner of each image. ISH for *RSPO4*, *BMP5*, *MSX1*, and *WIF1*: brown chromogen; IHC for CD10: red chromogen. Arrowheads in **f** and **g** indicated the co-labeled cells. FC fold change, ISH in situ RNA hybridization, IHC immunohistochemistry.

epithelium than normal skin epithelium (Supplementary Fig. 5b, c). We initially thought that *WNT6* expression is specific to the nail keratinocyte of polydactyl since structural anomaly of *WNT6* gene is linked to polydactyly syndrome[18]. However, *WNT6* was expressed in both the polydactyly samples and adult nail matrix suggesting *WNT6* expression on keratinocytes is nail-specific.

***LGR6* demarcated nail-specific stem cells**. Next, we tried to investigate the clusters harboring nail stem cells. We explored *LGR6* expression on nail-specific keratinocytes since *LGR6* has demonstrated to be a marker for nail stem cells in mice[19]. By comparing *LGR6* and *WNT6* expressions amongst the two nail-specific keratinocyte subpopulations, we found that they were limited to the Keratinocyte-1 cluster, implying that it was mainly isolated from nail matrix (Fig. 3d). Also, ISH studies showed *LGR6* was mainly expressed in the basal layer of nail matrix just above the *RSPO4* expressing mesenchyme (Fig. 3e–g). *LGR6* expression was strong in the basal layer of the proximal nail matrix. *WNT6* expression extended into higher levels of the nail matrix epithelium (Fig. 3h).

**Network analysis predicted interactions between onychofibroblasts and basal layer of nail matrix epithelium**. Localization of *LGR6* expression prompted us to analyze whether interactions between *LGR6* expressing basal keratinocytes and *RSPO4* positive onychofibroblasts in the subjacent mesenchyme exist. To assess the potential roles of *LGR6* and *RSPO4* in the epithelial–mesenchymal interactions of the nail unit, we used NichNet[20] to predict interactions between various receptors and ligands hypothesized in having roles in modulating nail development and regeneration. NicheNet predicted significant interactions between the *RSPO4* onychofibroblasts and *LGR6* at the basal layer of nail matrix epithelium (Fig. 4a). The interactions were further supported by a ligand-target link between *RSPO4* and the *LEF1* gene, a downstream target of *RSPO4* (Fig. 4b). To validate whether *RSPO4* and *LGR6* expression correlate with the activation of Wnt/β-catenin signaling, we performed IHC for β-catenin and LEF1 (Fig. 4c, d). We observed an increase of nuclear localization β-catenin in the nail matrix epithelium, indicating the activation of β-catenin signaling (Fig. 4c)[21]. Coincided with increased nuclear β-catenin, its transcriptional partner, LEF1 was also highly expressed in the nucleus of lower layers of the nail matrix (Fig. 4d).

Previously, based on histomorphologic and immunohistochemical studies, we proposed the terms onychodermis (nail-specific dermis located below the nail matrix and nail bed) and onychofibroblasts (fibroblast situated within the onychodermis)[6,7]. The proximal portion of the onychodermis was slightly separated from the undersurface of the nail matrix by a zone of connective tissue lacking CD10 expression. In our human nail scRNAseq data, many CD10-positive cells were RSPO4-positive, however, some RSPO4-positive cells were CD10-negative (Fig. 2b). ISH showed

RSPO4-positive cells in a zone of connective tissue immediately beneath the nail matrix. Considering the findings of our previous studies as well as this study, a zone of connective tissue containing RSPO4-positive cells right below the nail matrix should be included in the nail-specific dermis. Thus, we propose to expand the definition of the onychodermis containing onychofibroblasts to include this latter area, as delineated in Fig. 4e.

**Clinicopathologic and molecular characteristic of onychomatricoma**. Next, we searched for pathologic conditions arising from the specialized nail tissue we defined. Onychomatricoma is a tumor arising within or in the vicinity of the nail matrix (Fig. 5a–c). We hypothesized that onychomatricoma is arising from onychofibroblasts[22] and evaluated CD10 expression in onychomatricoma (Fig. 5f, h). We found that CD10 was strongly expressed in the mesenchymal part of the tumor. In addition, *RSPO4* was highly expressed in the fibroblast of onychomatricoma and an increased density of these fibroblasts compared to the normal nail unit, suggesting WNT signaling contributes to onychomatricoma histogenesis (Fig. 5g). To further investigate the involvement of WNT signaling in the pathogenesis of onychomatricoma, we tested the expression patterns of other WNT-related molecules. We found that *LGR6* was highly expressed at the basal layer of the overlying epithelial component (Fig. 5d). Also observed was *WNT6* expression on the epithelial component of the onychomatrichoma (Fig. 5e). *BMP5*, *MSX1*, and *WIF1* were expressed on the fibroblasts of onychomatricoma (Fig. 5j). Furthermore, IHC analyses for β-catenin and LEF1 showed strong nuclear β-catenin and LEF1 expressions in the epithelial cells of onychomatricoma (Fig. 5i). Together, our histological analyses suggest that WNT signaling is a key molecular characteristic of onychomatricoma.

**The functional importance of onychofibroblasts in the nail epithelial-mesenchymal interaction**. To understand the importance of RSPO4 in the nail matrix keratinocytes (NMKs), we sought to characterize transcriptomic changes induced by the RSPO4 treatment in cultured NMKs using RNAseq analysis. Bioinformatic analyses showed that in the comparison with control NMKs, 26 differentially expressed genes (DEGs) were identified in the NMKs treated with RSPO4 1000 ng/ml, with 18 upregulated and 8 downregulated genes ($P$ value < 0.05 and absolute fold change > 2; Fig. 6a; Supplementary Table 3; Supplementary Data 2). The six replicates of RSPO4 1000 ng/ml treated, RSPO4 200 ng/ml treated, and control NMKs were analyzed by a hierarchical clustering and we found that RSPO4 treated groups showed the upregulation of WNT signaling associated genes including *LGR6* and *FOXQ1* (Fig. 6b, c)[23]. By comparing the transcriptomes of onychomatricoma with the nail matrix biopsy sample, we observed similar trends of the changes of WNT signaling associated genes, *LGR6* and *FOXQ1*. These results suggest that epithelial hyperplasia in onychomatricoma

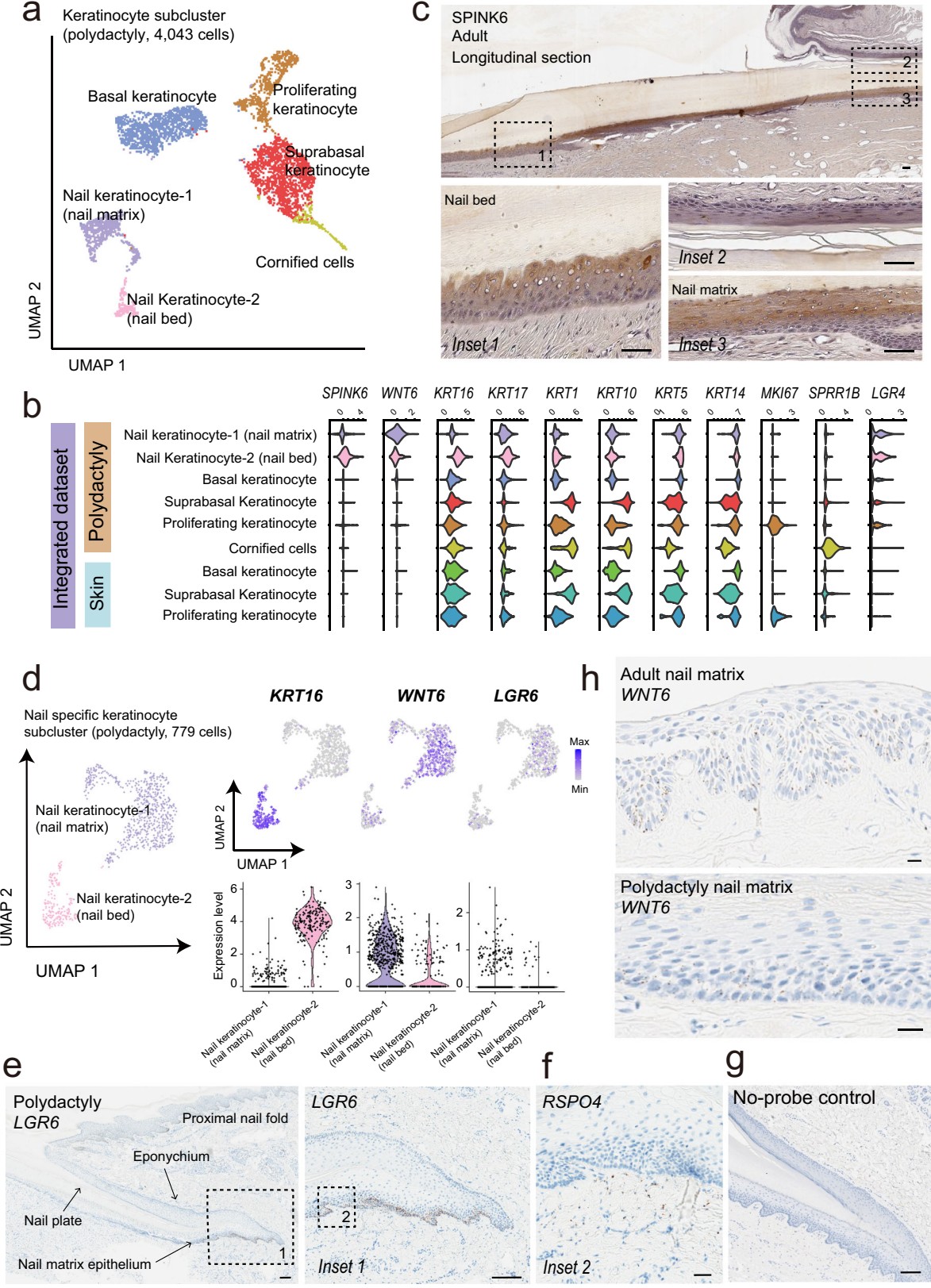

was potentially induced by overexpression of *RSPO4* in its mesenchyme.

In addition, we treated cultured NMKs with BMP-5 or WIF-1, which are highly expressed by onychofibroblasts (Supplementary Fig. 8). RNAseq analysis identified 181 DEGs for BMP5 100 ng/ml treated NMKs versus control NMKs, and 153 DEGs for WIF-1 1000 ng/ml treated NMKs versus control NMKs (Supplementary Fig. 8). There was no significant difference in *LGR6* expression.

**Comparison of human nail unit with human hair follicle**. To gain a deeper understanding of nail biology, we compared our

**Fig. 3 Subclustering of keratinocytes derived from polydactyly identifies the nail-specific cell populations. a** A UMAP of keratinocyte reveals six distinct clusters. **b** The violin plot shows an expression of nail-specific markers (*SPINK6* and *WNT6*) in keratinocytes from the publicly available skin dataset and polydactyly dataset. **c** Immunohistochemical staining reveals a strong expression of SPINK6 in the nail epithelium. Scale bar = 50 μm. **d** Feature and violin plot represent *KRT16* expression on nail keratinocyte-2 (nail bed) and, *WNT6* and *LGR6* on nail keratinocyte-1 (nail matrix). **e** ISH of *LGR6* on polydactyly tissue (with a magnified view of nail matrix indicated by dashed-black box on the right). Scale bar = 100 μm. **f** Additional ISH of boxed area in **e** (inset 2) using mRNA probes for *RSPO4*. *RSPO4* expression is subjacent to *LGR6*+ basal layer of nail matrix epithelium. Scale bar = 20 μm. **g** No-probe control. Scale bar = 100 μm. **h** ISH of *WNT6* on adult nail matrix tissue and polydactyly tissue. Scale bar = 20 μm. ISH in situ RNA hybridization.

nail unit scRNAseq data with previously reported human hair unit scRNAseq data[11,24] (Supplementary Figs. 6b and 9a; Supplementary Table 2). Integrated data showed human nail unit shared transcriptomes in common with the human hair unit (Supplementary Fig. 9b). However, we were not able to define RSPO4+ onychofibroblasts or SPINK6+ nail epithelium analogs in the clusters of public human hair datasets. This could be, in some parts explained by few mesenchymal cells included in the public datasets. Since we defined the presence of onychofibroblasts through several morphological and immunohistochemical studies[6,7,13] and we now demonstrate *RSPO4* expression in onychofibroblasts (Supplementary Fig. 9c), we conducted *RSPO4* ISH on human hair-bearing skin to identify mesenchymal populations analogous to the onychofibroblasts. Importantly, the follicular dermal papilla cells showed high expression of *RSPO4* (Supplementary Fig. 9d). *LGR5* expression was found at the closest proximity to the *RSPO4* expressing follicular dermal papilla, compared to the other two remaining RSPO4 receptors, *LGR4* and *LGR6* (Supplementary Fig. 9e, f).

## Discussion

In this study we revealed transcriptome profiles of human nail units by polydactyly samples at a single-cell resolution. By sub clustering fibroblasts and keratinocytes respectively we identified one nail-specific mesenchymal cell population and two nail-specific epithelial cell populations.

Nail-specific mesenchymal cell population (onychofibroblast cluster) which were characterized by *MME* (CD10) expression demonstrated high expression of *RSPO4*. The expression of *RSPO4* in the mesenchymal cells below the nail matrix and nail bed in polydactyly and adult nail unit was confirmed by ISH. *RSPO4* is a member of the R-spondin family of secreted protein which is involved in Wnt/β-catenin signaling[25]. Previously, *RSPO4* expression was localized to developing mouse nail mesenchyme, suggesting a crucial role in nail morphogenesis[26]. In addition, a gene encoding RSPO4 in humans is mutated in inherited anonychia or hyponychia[27]. Thus, our data demonstrate that *RSPO4* expression in onychofibroblasts is crucial for human nail development and maintenance. We also found additional genes that were strongly and specifically expressed in the onychofibroblast cluster including *MSX1*, *TWIST1*, *CRABP1*, *WIF1*, and *BMP5* (Fig. 2c). Mice with a homozygous deletion of *MSX1* exhibited a complete cleft palate and failure of tooth development[28]. In humans, *MSX1* mutation is associated with orofacial clefting and tooth agenesis[29]. A nonsense mutation in *MSX1* causes Witkop syndrome, also known as tooth and nail syndrome[30]. *MSX1* participates in the regulation of WNT signaling by induction of WNT antagonist *DKK1*, *DKK2*, *DKK3*, and *SFRP1*[31]. We also observed the expression of *DKK1* in chondrocytes (Supplementary Fig. 2). *TWIST1* is recently spotlighted as a regulator of cancer-associated fibroblast[32]. This basic helix–loop–helix transcription factor is also essential for the development of mesodermally derived tissues including muscle and bone. *TWIST1* mediated fibroblast activation implicates that onychofibroblasts are metabolically active. *CRABP1* is known to be expressed on the dermal papilla of developing hair[33]. Murine

scRNAseq results dissecting fibroblast heterogeneity in skin demonstrated CRABP1+ fibroblasts were enriched subjacent to the epidermis, while CRABP1− fibroblasts were primarily localized to the deeper dermis[34]. These reports are comparable with our findings in the nail unit, with positive *CRABP1* gene expression in onychofibroblasts and our localization of onychofibroblasts also in the juxta-epithelial areas. *WIF1* is known as WNT signaling regulator and a potential cause of a Nail-Patella-like disorder[35] and *BMP5* is reported to regulate murine keratinocyte stem cells[36]. Taken together, high expression of *RSPO4* and expression of other genes related to nail abnormalities in onychofibroblasts suggest that onychofibroblasts play an important role in nail formation and growth through epithelial-mesenchymal interactions.

Two nail-specific epithelial cell populations demonstrated high expression of *SPINK6* and *WNT6*. The expression of *SPINK6* and *WNT6* in the nail-specific epithelial cells was confirmed respectively by IHC and ISH. Previously, *SPINK6* was found to be expressed in the stratum granulosum of human skin and in the skin appendages, and it was identified as a selective inhibitor of human Kallikrein-related peptidases in human skin[17,37]. In addition, *SPINK6* expression was found in the claw region of mouse[38]. However, in quantitative RT-PCR analysis of our study *SPINK6* expression in the epidermis was too low compared with the nail matrix. In addition, by IHC SPINK6 expression was not found in the epidermis. Considering SPINK family's roles as a protease inhibitor, *SPINK6* might have a role in maintaining epithelial barrier in the nail unit. In the past, Wnt6 expression was reported in the developing mouse skin[39]. This is the first report of *WNT6* expression in the nail matrix and nail bed of the nail unit. In our study, we found that *LGR6* was mainly expressed in the nail matrix that overlies the onychofibroblasts. *LGR6* was strongly expressed in the basal layer of proximal nail matrix. Previously, it was reported that nail stem cells reside in the proximal nail matrix of mice[40]. In addition, *LGR6* were expressed in the nail matrix and demonstrated to be a marker for nail stem cells in mice[19].

Further network analysis using NicheNet predicted significant interactions between *RSPO4* secreted by onychofibroblasts with *LGR6* positive nail matrix epithelium. A link was also found between *RSPO4* and the *LEF1* gene, a downstream target of *RSPO4*. These findings suggest interactions between *RSPO4* secreted by onychofibroblasts with *LGR6* in the nail matrix, possibly resulting in activation of the Wnt/β-catenin signaling. Indeed, functional activation of β-catenin-LEF1 was confirmed in the nail matrix just above the *RSPO4* expressing mesenchyme (Fig. 4c, d). These results are consistent with findings in previous murine studies evaluating mechanisms underlying epithelial-mesenchymal interactions in the nail unit[41]. To identify genes regulated by onychofibroblasts in nail matrix epithelium, we profiled the gene expression changes following RSPO4, WIF1, and BMP5 treatment in cultured NMKs. We hypothesized that RSPO4 protein might induce activation of *LGR6* positive nail matrix epithelium. Strikingly, RSPO4 treatment induced the increase of *LGR6* expression, whereas *LGR6* expression was not influenced by BMP5 or WIF1 treatment. Notably, we also

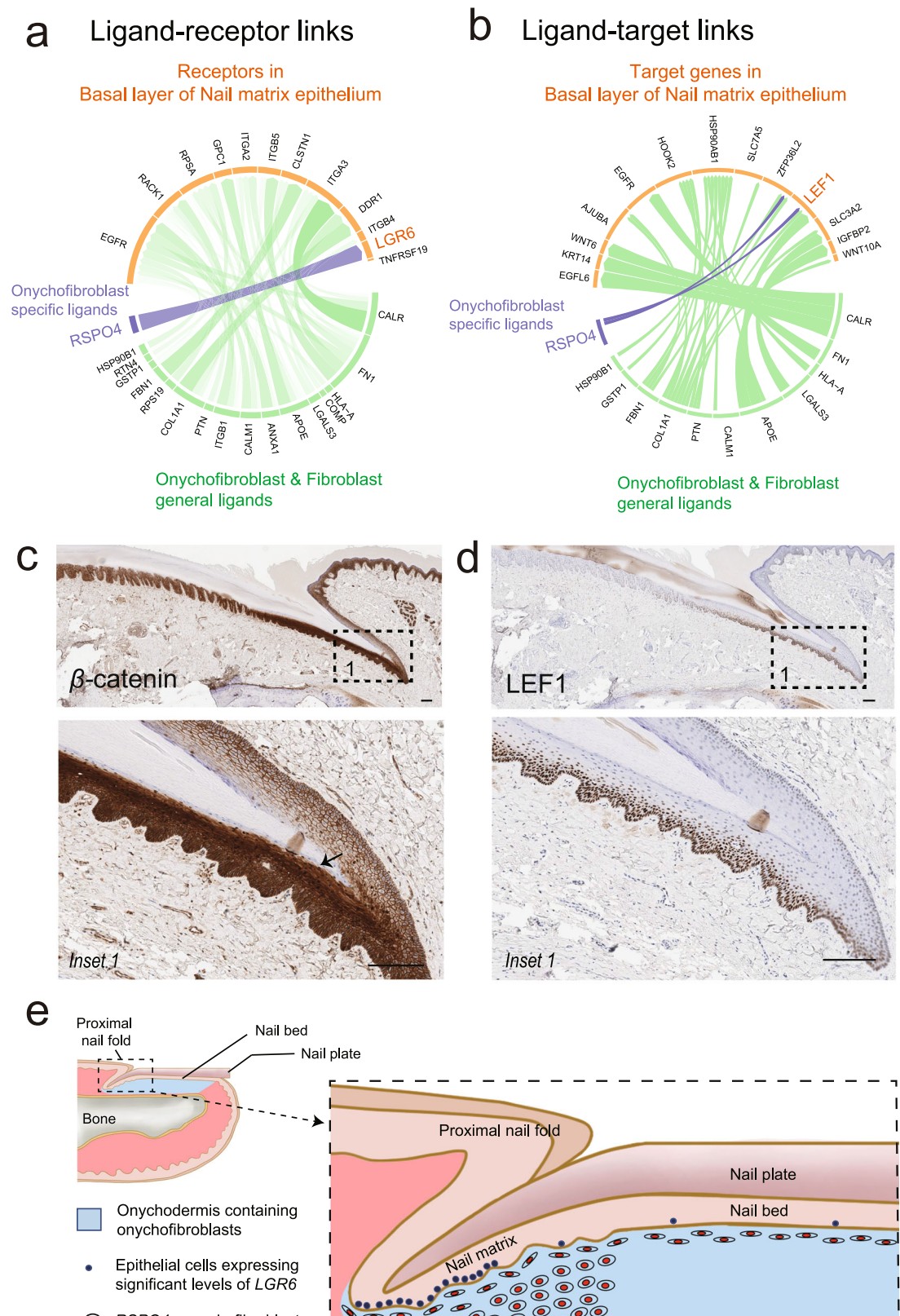

observed overexpression of *FOXQ1* which is considered as the marker for WNT signaling activation in solid tumors[23]. These results highlight the potential role of RSPO4 in the nail development through WNT signaling.

Onychomatricoma is a rare benign tumor of the nail unit, usually located at or around the nail matrix[42]. Although its name

implies a tumor of the nail matrix epithelium, it is a fibroepithelial tumor with both epithelial and mesenchymal elements (Fig. 5), and thus the concept of epithelial onychogenic tumor with onychogenic mesenchyme was introduced[43]. Indeed, we observed very strong expression of *RSPO4* and *LGR6* in the mesenchymal cells and epithelial components of the tumor,

**Fig. 4 Network analysis predicted interactions between onychofibroblasts and basal layer of nail matrix epithelium. Schematic diagram of human nail unit was described. a** NicheNet analysis between ligands in fibroblast clusters and receptors in the basal layer of nail matrix epithelium. Greenline represents the connections involving onychofibroblasts and non-onychofibroblasts. Blue line represents the connection specifically involving onychofibroblasts. *RSPO4* in onychofibroblasts specifically connected to *LGR6*. **b** Network analysis reveals interaction between ligands in onychofibroblast clusters and WNT pathway target genes. **c**, **d** IHC showing expression of β-catenin and LEF1. Cells with strong nuclear β-catenin staining can be observed in the suprabasal layers of proximal nail matrix epithelium (arrow). **e** Schematic outline illustrating the location of *RSPO4*+onychofibroblasts in the onychodermis and *LGR6*+ keratinocytes in the human nail unit. Scale bar = 100 μm. ISH in situ RNA hybridization, IHC immunohistochemistry.

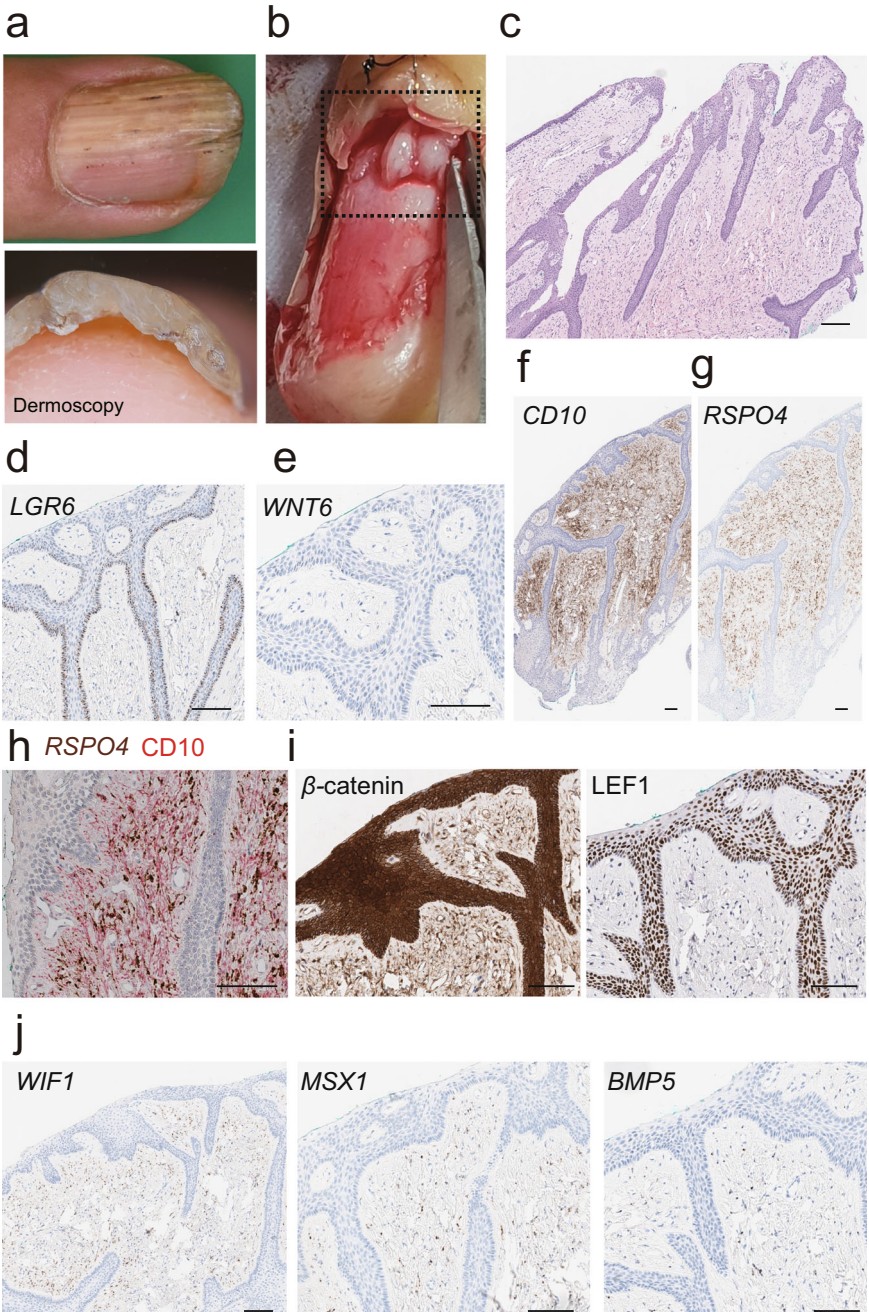

**Fig. 5 Clinico-histopathologic findings of onychomatricoma. a** Clinical images of onychomatricoma. **b** Surgical pictures of onychomatricoma. **c** Representative image of H&E from surgical resection. **d**, **e** ISH of *LGR6* and *WNT6* on onychomatricoma. **f** IHC of CD10. **g** ISH of *RSPO4*. **h** Dual *RSPO4* ISH and CD10 IHC. **i** Representative IHC showing expression of β-catenin (left) and LEF1 (right), markers for Wnt/β-catenin signaling. Cells with nuclear β-catenin staining are found in the epithelial region of onychomatricoma. **j** ISH of *WIF1*, *MSX1*, and *BMP5* on onychomatricoma. Scale bar = 100 μm. ISH in situ RNA hybridization, IHC immunohistochemistry.

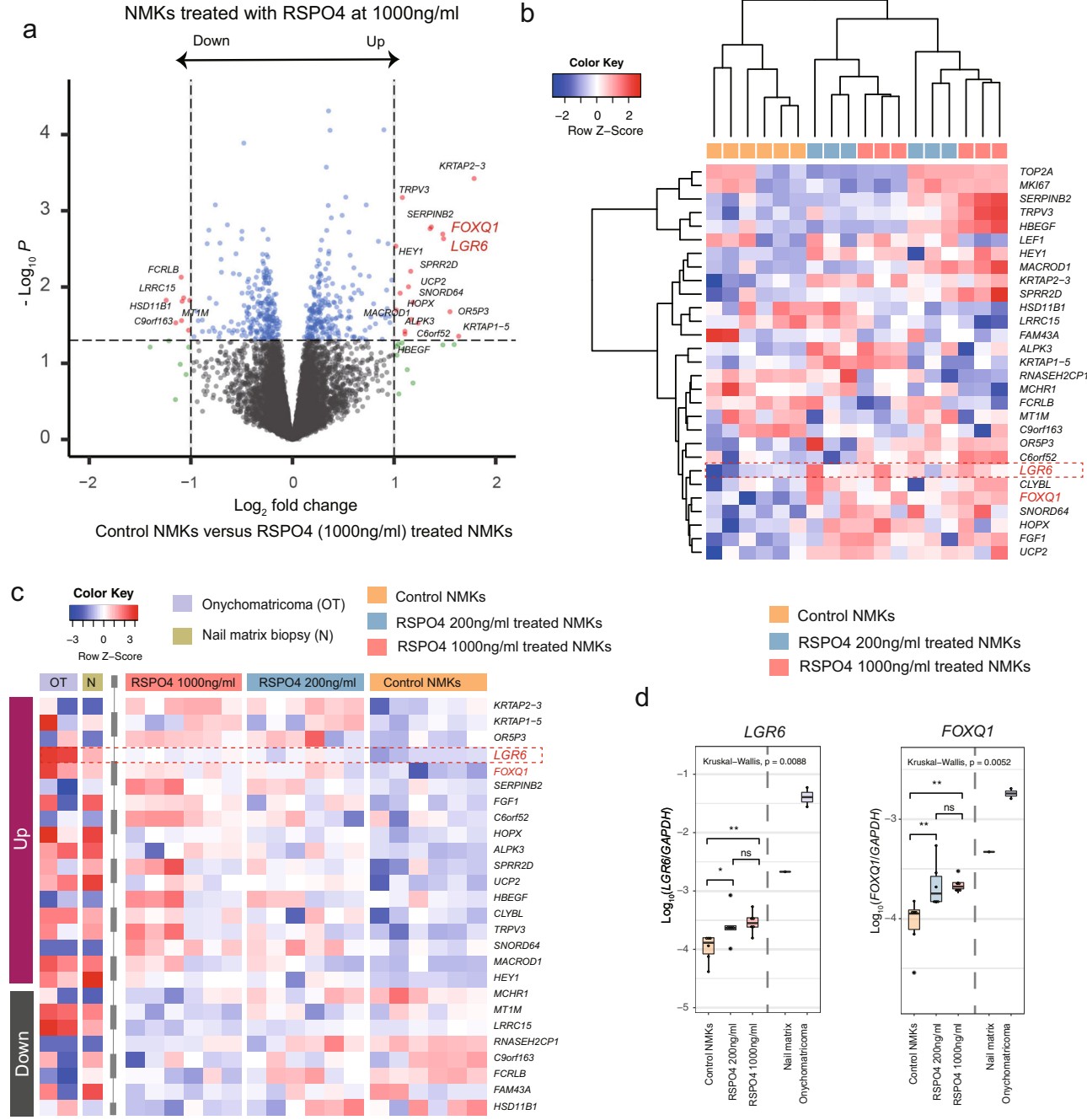

**Fig. 6 RSPO4 regulated epithelial *LGR6* expression in nail matrix keratinocytes (NMKs). a** Volcano plot depicting differentially expressed genes (DEGs) for RSPO4 (1000 ng/ml) treated and control NMKs. A total of 26 DEGs were identified, including *LGR6* and *FOXQ1*. **b** Heatmap of DEGs, as determined in Fig. 6a. DEGs defined by *P* value < 0.05 and fold change >2 cutoffs. *LEF1*, *MKI67*, and *TOP2A* were also used for the heatmap. The 6 replicates RSPO4 1000 ng/ml treated, 6 replicates RSPO4 200 ng/ml treated, and 6 control NMKs were used for hierarchical clustering. WNT signaling-associated genes are shown in red. **c** Heatmap showing mRNA expression (*Z* score) of the upregulated and downregulated genes, as determined in Fig. 6a. The two onychomatricoma and one nail matrix biopsy sample were also used for analysis. **d** Boxplots of WNT signaling associated gene expression (log$_{10}$ value of mRNA expression relative to *GAPDH* expression) under the indicated culture condition. The onychomatricoma and nail matrix biopsy samples were also included. Mann–Whitney *U* test was used to assess statistical significance. *$P < 0.05$; **$P < 0.01$; ns not significant. NMKs nail matrix keratinocytes, DEGs differentially expressed genes.

respectively. Of note, we also observed strong nuclear β-catenin staining, implying translocation of cytosolic-β-catenin into the nucleus, and high expression of LEF1, a transcriptional coactivator of β-catenin, in the epithelial cells of onychomatricoma[44]. Previously, we suggested that onychofibroblasts may be involved in the pathogenesis of the onychomatricoma[22]. Interestingly, we observed a significant upregulation of *LGR6* in the RSPO4 treated NMKs (Fig. 6c). The expression pattern observed in this study suggests a possible link between *RSPO4* positive onychofibroblasts and overexpression of *LGR6*, which could lead to dysregulation of Wnt/β-catenin signaling in onychomatricoma. These findings are in line with previous studies that RSPO proteins could lead to a significant epithelial proliferation in the gastrointestinal tract[45,46]. Collectively, we suggest onychomatricoma is

a nail-specific tumor with mesenchymal origination from onychofibroblasts, rather than a nail matrix epithelium-limited process, and propose an alternate term onychofibroblastoma for this entity.

As the nail unit and hair follicle share numerous similarities[3,4], we hypothesized that the onychofibroblasts could be the nail counterpart of follicular dermal papilla. We demonstrated that the follicular dermal papilla cells showed high expression of *RSPO4* suggesting the possibility they may be counterpart of onychofibroblasts. However, there appears to be a spatial gap between cells expressing *RSPO4* and cells harboring its putative receptors (*LGR4–6*). Additional studies are needed to elucidate whether and how *RSPO4* secreted by dermal papilla interact with ligands in the hair epithelium.

In summary, through integrated analysis of molecular data and IHC staining, we demonstrated nail-specific mesenchymal and epithelial cell populations, which are characterized by *RSPO4* and *SPINK6*, respectively in human nail unit. We identified that *RSPO4*+ onychofibroblasts are situated at close proximity with the *LGR6*+ nail matrix, which lead to WNT/β-catenin activation. We demonstrated evidence of analogous findings in the mesenchymal and epithelial components of onychomatrichoma, a pathologic hyperproliferative state involving the nail matrix, and we conclude that onychomatricoma is a tumor which derives substantial portion of its origination from onychofibroblasts. Altogether, this study highlights the importance of the interaction between onychofibroblasts and *LGR6*+ nail matrix epithelium, which contribute the nail formation and growth via WNT/β-catenin signaling pathway. Although further analysis will be needed to identify *RSPO4*-mediated downstream signaling responsible for the precise controlling of the nail homeostasis, our findings shed light on molecular basis of the nail biology.

## Methods

**Patient cohort(s) and sample preparation**. All samples were collected under Institutional Review Board (IRB)-approved consent (IRB number: SMC 2017-10-137). All patients provided written informed consent. Extra-digits from 4 patients (6–12 months) were delivered to the dermatologic clinic immediately after the digit removal. All samples containing nail mesenchyme and epithelium were dissected, minced and dissociated within 3 h of digit removal. Specimens were transferred to freshly prepared dissociation solution composed of 120 µl of Liberase TL (2 mg/ml; Sigma Aldrich) and 840 µl phosphate-buffered saline (PBS) and incubated at 37 °C for 45 min. The cells were collected through 70-µm cell strainer (#352340, Corning) and stored on ice. The tissue was transferred into dissociation solution for a second round of dissociation as mentioned above followed by dissociation in Trypsin solution (350 µl PBS, 50 µl 0.25% Trypsin). Cells were washed once and re-suspended in 50–200 µl of freshly prepared PBS–bovine serum albumin (BSA) (1× PBS and 0.04% BSA). Cell viability was assessed using LUNA-FL dual fluorescence cell counter (Logos biosysteomics).

For the onychomatricoma samples, tissues were obtained from two patients after surgical resections. The diagnosis was confirmed on the basis of pathologic review and clinical examination.

scRNAseq data for the skin and hair dataset were obtained from two independent published cohort of five human skin samples[11] and two human scalp samples[24]. Additional details regarding these patient cohorts are presented in Supplementary Table 2.

**Single-cell capture, library preparation, and sequencing**. Single-cell dissociates were loaded into the Chromium system (10× Genomics) targeting 5000 cells. The Chromium Single Cell 3′ Reagent Kit V2 (10× Genomics) was used to generate scRNAseq libraries, according to the manufacturer's instructions. Barcoded sequencing libraries were generated using Chromium Single Cell 3′ v2 Reagent Kits and sequenced across by a HiSeq 4000 platform targeting 25,000 reads per cell.

**scRNAseq data align and quality control filtering**. Sequencing data were aligned to the human reference genome (GRCh38) and processed using the CellRanger 3.1.0 pipeline (10× Genomics). The raw gene expression matrix from the Cell-Ranger pipeline was filtered, normalized using the Seurat package version 3.1.1 in R version 3.6.0 software (R Foundation for Statistical Computing, Vienna, Austria.) and selected according to the following criteria: cells with >500 unique molecular identifier (UMI) counts; and <6500 genes; and <30% of mitochondrial gene expression in UMI counts. In order to integrate multiple single-cell datasets and

correct for batch effects, we performed the standard integration protocol described in Seurat v3. Briefly, we performed standard preprocessing (log-normalization) and identify 2000 more variable genes per sample. We then used the function FindIntegrationAnchors() implemented in Seurat v3 and 40 canonical correlation analysis dimensions to identify the integration anchors between our datasets. A total of 11,541 remaining cells were enrolled in the final analysis.

**Clustering and single-cell RNA sequencing analysis**. The integrated data were used for graph-based clustering and visualization with Seurat R package. The 2000 most variable genes of the integrated dataset were used as input. ScaleData() function and RunPCA() function were used for scaling and principal component analysis (PCA), respectively. We clustered the cells with the FindNeighbors() and FindCluster() functions. The resolution and PCA dimensions were selected variably per sample origin; resolution 0.6 and 'dims = 1:40' for polydactyly samples, resolution 0.6 and 'dim = 1:20' for public hair graft samples[24], and resolution 0.4 and 'dims =1:20' for public skin samples[11]. After unsupervised clustering, we ran RunUMAP() with 40 PCA dimensions to visualize a multi-dimensional dataset. Differentially expressed genes (DEGs) for each cluster were identified using FindAllMarkers() function with default parameters. To determine cell types, we compared top DEGs between each cluster using cell type-specific markers known from a literature review (Supplementary Fig. 2).

For the second-level clustering, we isolated the keratinocyte cluster and fibroblast cluster in the broad cell type UMAP, respectively. We further subclustered these cell types by reapplying the FindNeighbors() and FindCluster() functions at resolution 0.6 with 20 PCA dimensions. Average log-normalized expression profiles for keratinocyte-associated genes were calculated using the AverageExpression() implemented in Seurat packages. For the gene ontology (GO) analyses, lists of genes differentially expressed between fibroblast (1–3) and onychofibroblast clusters were calculated using FindMarkers () with default parameters. These genes were further selected with a *P* value < 0.05 and average log (fold change) > 0.4. GO analysis was performed with Metascape web tool (www.metascape.org)[47].

**Ligand-receptor interaction analysis**. To investigate the epithelial-mesenchymal interactions potentially contributing to nail development and regeneration, we employed the ligand–receptor interaction tool NicheNet[20]. *LGR6*+ nail matrix cells (basal layer of the nail matrix) were isolated from the nail keratinocyte-1 (nail matrix) cluster using *LGR6* expression level. To determine genes expressed in each cluster, we used the definition used by of Puram et al.[48]. We defined the gene set of interest as the genes differentially expressed within interacting cell population. FindAllMarkers() function with default parameters was used to calculate the DEGs and these genes were further selected with a *P* value < 0.01 and average log(fold change) > 0.4. Ligand activities were predicted using the predict_ligand_activities() function implemented in NicheNet, and the top 30 ligands were selected by Pearson correlation coefficient. The upper 20% of the ligand-target links according to the regulatory potential scores were selected to be visualized in circos plot to avoid making a circus plot with too many ligand-target links.

**RNA in situ hybridization (ISH)**. Total of six independent polydactyly, five nail biopsy, one onychomatricoma and two hair bearing scalp specimens were included in this study. ISH for mRNA expression was performed on three micrometers-thick formalin-fixed paraffin-embedded of tissue samples. Staining was performed using the Leica Bond Rx autostainer with probes for RSPO4 (NM_001029871.3), MSX1 (NM_002448.3), BMP5 (NM_021073.2), LGR6 (NM_001017404.1), and LGR5 (NM_003667.2). All probes used for ISH were obtained from Advanced Cell Diagnostics (Newark, CA). Probes were detected using the RNAscope 2.5 LS Reagent Brown Kit (cat# 322100; Advanced Cell Diagnostics).

**Immunohistochemical staining**. Samples were sectioned longitudinally parallel to the long axis of the nail plate or transversally. Hematoxylin- and eosin-(H&E) stained slides of the specimens were reviewed to determine the most comprehensive section. Immunohistochemical staining was done using a monoclonal antibody against CD10 (clone 56C6; Novocastra, Newcastle, UK), SPINK6 (ab110830; Abcam, Cambridge, UK), β-catenin (14; Cell Marque, CA, USA) or LEF1 (ab137872; Abcam, Cambridge, UK). For IHC, tissue sections were deparaffinized with xylene and rehydrated with ethanol. To block endogenous peroxidase in the tissue sections, the slides were treated with hydrogen peroxide in methanol. Sections were incubated in a blocking solution (Protein block, DAKO) to prevent nonspecific antibody binding. The sections were then incubated with primary antibody for a few hours at room temperature in a humid chamber. The slides were washed in phosphate buffered saline, followed by antigen detection using DAKO EnVision System and diaminobenzidine or Fast Red for visualization. The slides were counterstained with hematoxylin solution.

**Microarray**. Two polydactyly samples were used for this experiment. Epithelial tissue was separated from dermal tissue after dispase treatment. Then nail epithelium and epidermis were obtained respectively. RNA microarray analysis was performed to measure gene expression levels. Expression profiling analysis was performed using Agilent Oligo Microarray Kit 8x60K according to the Agilent

One-Color Microarray-based Gene Expression Analysis Protocol (Agilent Technologies). Statistical significance of the expression data were determined using LPE test and fold change in which the null hypothesis was that no difference exists among groups. False discovery rate was controlled by adjusting p value using Benjamini–Hochberg algorithm.

**Quantitative real-time RT-PCR.** Four polydactyly and two cadaveric samples were used in this study. Nail epithelium and epidermis were obtained respectively. Total RNA was isolated using TRIzol reagent (Invitrogen, Carlsbad, CA). SPINK6 gene-specific PCR primers (forward primer, 5′-ACC TCA GCT GGA CAA AGC AG-3′; and reverse primer, 5′-TGG CAA GTC ACC AAG AAA CA-3′) were designed following the previous study published[17]. The expression levels of SPINK6 gene were analyzed by the number of copies per copy of GADPH.

**Nail matrix cell culture and treatment of RSPO4, BMP-5, and WIF-1.** Primary NMK was harvested from nail matrix biopsy specimens. Briefly, after removing the lower portion from nail matrix tissue, primary NMKs were harvested from nail matrix biopsy specimens. NMKs were isolated after 0.25% trypsin–EDTA treatment for 30 min and cultured in Keratinocyte Growth Media (KGM, Lonza) and maintained at 37 °C with 5% $CO_2$ as described previously[49]. We checked no fibroblast growth under an inverted microscope. A total of 100,000 cells which are passage five or six were plated in triplicate in six-well plates. Recombinant human RSPO4, BMP-5, or WIF-1 (R&D systems) were treated, respectively in KGM without rhEGF. After 48 h of culture, NMK was treated with 1 ml of Trizol to isolate RNA for further bulk RNAseq experiments.

**RNA isolation and bulk RNA-seq analysis.** Total RNA was extracted and the libraries were prepared using the TruSeq Stranded Total RNA Sample Prep Kit with Ribo-Zero H/M/R (Illumina Technologies., San Diego, CA, USA). The concentration and integrity of extracted RNA was evaluated by Nanodrop 8000 UV-Vis spectrometer (NanoDrop Technologies Inc., Wilmington, DE, USA). Total 1 μg of RNA molecules were pooled for cDNA synthesis and these cDNA libraries were qualified with the Agilent 2100 BioAnalyzer (Agilent Technologies, Santa Clara, CA, USA). After cluster amplification of denatured templated, the sequencing of each library was performed as150-bp paired-end using Illumina Novaseq6000 (Illumina Technologies., San Diego, CA, USA).

The RNA-sequencing reads were mapped to the human reference genome (hg19) with STAR and gene expression level was quantified using RNA-seq by expectation–maximization[50]. Normalized gene counts were transformed to log2-counts per million (logCPM) with the R package, edgeR[51]. DEGs were identified using the R package, edgeR[51].

**Statistics and reproducibility.** All statistical analyses were performed using R3.6.0 software (R Foundation for Statistical Computing, Vienna, Austria). Statistical analyses of the scRNA-seq data (n = 4, see Supplementary Table 1) were performed using the CellRanger and Seurat packages in R. The Mann-Whitney U test and paired Wilcoxon test was used for comparison of differences in means using R. A P value less than 0.05 was considered to indicate a statistically significant difference for all comparisons. All reported P values were two-sided. As stated above, IHC experiments and RNA ISH were performed in 1–6 replicates per genes.

**Reporting summary.** Further information on research design is available in the Nature Research Reporting Summary linked to this article.

## Data availability

The scRNAseq data that support the findings of this study has been deposited in the the Gene Expression Omnibus (GEO) under accession code GSE158970. The scRNA seq data from previously published cohorts are available at the GEO under the following accession number: GSE129611 and GSE130973. Any other data are available from the corresponding author upon reasonable request.

## Code availability

For the Seurat computational pipeline, we used seurat clustering standard work flow (https://satijalab.org/seurat/vignettes.html). The codes generated during this study are available at Github repository (https://github.com/SMC-Derma/scRNAseq_NailMatrix).

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

## Acknowledgements
This work was supported by NRF-2018R1D1A1A02086103.

## Author contributions
D.L. conceived and designed the study. J.S.S., K.-H.L., J.-H.P., and K.-T.J. contributed to the sample acquisition and patient recruitment. H.J.K. and J.-H.P. processed the samples. J.H.S. and D.L. analyzed and interpreted the data. J.-M.Y., E.J.K., H.T.S., W.-Y.P., H.Y., H.-S.J., and J.H.L. edited the paper. H.J.K., J.H.S., and D.L. wrote the paper with the input from other authors. All authors read and approved the final paper.

## Competing interests
The authors declare no competing interests.
