## [Peer Review File · Communications Biology]

REVIEWERS' COMMENTS:

Reviewer #1 Review

The study is based on the hypothesis that CD10 expressing fibroblasts in the digit are important for nail growth and development through an epithelial-mesenchymal signaling relationship. While this is a logical line of investigation, the authors have not convincingly addressed this hypothesis, and instead present a descriptive expression/correlation analysis of the human digit fibroblasts and nail epithelium. The newly added functional analysis that is presented (RSPO4 nail matrix keratinocyte analysis) does not provide new molecular insight into the nail, as RSPO4 has already been described in the nail and is known to agonize WNT signaling; WNT signaling is also already described in the mouse nail epithelium.

Specific comments:

The authors define fibroblast cluster 4 as “onychofibroblasts” based on CD10 expression, however there is some expression of CD10 in clusters 1-3 as well, suggesting that either the clustering is underpowered and/or CD10 is potentially expressed in other fibroblastic cell types. The authors propose there are two different CD10 cell-types based on morphology, but this is a hypothesis and not explicitly explored (Sfig3).

Along the same lines, most of the featureplots for onychofibroblast markers (Sfig 4) show broader expression than just cluster 4. While these could be onychofibroblastic-specific markers, the alternate hypothesis is that these genes are heterogeneously expressed within the digit dermis. The co-labeled sections in 2f could be interpreted as heterogeneously expressed factors that are stochastically co-labeled with CD10+ cells. The colorimetric in situ make it hard to appreciate the co-localization; quantification of singly and doubly labeled cells, as well as arrows would help the reader.

The authors state that aberrant expression of Spink6 was found in 20 dysmorphic nails but the data is not presented. This data is key to the authors' conclusion, thus it should be presented or the statement should be removed.

Insufficient methods have been provided for the newly added nail matrix keratinocyte experiments. How were fibroblasts excluded from the culture? How were the doses of recombinant proteins chosen (the concentrations of RSPO4 seem high)?

Reviewer #2 Review

I have carefully examined the revised paper. The authors have improved expression analyses on human nail tissue sections in this revision in response to my comments. I consider provided information from single cell RNA seq analyses coupled with histochemical validation is unique and valuable, particularly considering the rare availability of these human nail tissue samples. However, the manuscript still remains entirely descriptive without an experiment to functionally validate their theory. As the authors can culture human nail matrix cells and fibroblasts, there are multiple approaches with which they can at least partly validate their theory to concretely demonstrate a novel example of epidermal-fibroblast interaction. In my view, the paper would be beneficial to a specialized scientific community in dermatopathology/dermatology.

Reviewer #3 Review

The authors addressed most of the concerns brought up in the previous review. Previously, they

identified a group of RSPO4+ onychofibroblasts by scRNA-seq but didn't address the functional importance of this group of cells. Now they included the observation that RSPO4+ cells directly underlie Lgr6+ NMKs (nail matrix keratinocyte) in which B-cat and Lef1 are also upregulated. They thus hypothesized that RSPO4+ onychofibroblasts induce Wnt signaling in Lgr6 cells. To validate the hypothesis, they treated NMK cell culture with RSPO4 and conducted RNA-seq. The results showed that RSPO4-treated NMKs exhibited significantly higher level of Lgr6 and FOXQ1 (Wnt marker in solid tumor) than control NMKs. The result not only suggests that RSPO4 secreted by onychofibroblasts is important for Lgr6 level in NMKs, but also that overexpression of RSPO4 might be linked to onychomatricoma, a form of benign nail tumor. In general, there were improved annotations and insets that make their working model clearer to the reader. This study provides a helpful study and resource in a field that is less well-characterized than other skin appendages and can help aid further functional investigations.

1) In line 228 of text 'dose-dependent' effect of RSPO4 was mentioned, although it's not clear to me how figure 6d would suggest that the Lgr6 level is different between 200ng/ml and 1000 ng/ml treatment groups.

2) Lef1 is not shown in Figure 6. Can the authors show Lef1 status since it is the Wnt marker used in Figure 4 and 5 and is a more well-established Wnt target gene.

3) proliferation marker/assays can be used to show that RSPO4 treatment induces cancer-like characteristics in NMKs.

4) The marker HEY1 is highlighted in figure 6 but not explained in text.

Response to the reviewers

We thank the reviewers for the detailed comments and suggestions, which led to significant improvements. In the text below, reviewer comments are given, followed by our response (text with blue color)

REVIEWERS' COMMENTS:

Reviewer #1 Review

The study is based on the hypothesis that CD10 expressing fibroblasts in the digit are important for nail growth and development through an epithelial-mesenchymal signaling relationship. While this is a logical line of investigation, the authors have not convincingly addressed this hypothesis, and instead present a descriptive expression/correlation analysis of the human digit fibroblasts and nail epithelium. The newly added functional analysis that is presented (RSPO4 nail matrix keratinocyte analysis) does not provide new molecular insight into the nail, as RSPO4 has already been described in the nail and is known to agonize WNT signaling; WNT signaling is also already described in the mouse nail epithelium.

> As mentioned in the first revision, the newly added functional analysis (*RSPO4* nail matrix keratinocyte analysis) provided that *RSPO4* induces Wnt signaling in human nail matrix keratinocytes, especially significantly higher level of *LGR6*. In addition, RNA sequencing data in onychomatricoma showing similar trends of the changes of WNT signaling associated genes, *LGR6* and *FOXQ1*, suggest mechanistic insights through epithelial-mesenchymal interactions in the nail unit.

Previously, in situ hybridization of mouse embryos revealed RSPO4 expression at the sites of nail development. However, RSPO4 and its activation of WNT signaling were not reported in human nail tissue (infant as well as adult nail tissue). We demonstrated RSPO4 expression in the mesenchymal cells (onychofibroblasts) only below the nail matrix and nailbed of human nail unit, but not digit dermis. We also demonstrated strong expression of RSPO4 in the fibroblasts of onychomatricoma.

Specific comments:

COMMENT #1-1

The authors define fibroblast cluster 4 as “onychofibroblasts” based on CD10 expression,

however there is some expression of CD10 in clusters 1-3 as well, suggesting that either the clustering is underpowered and/or CD10 is potentially expressed in other fibroblastic cell types. The authors propose there are two different CD10 cell-types based on morphology, but this is a hypothesis and not explicitly explored (Sfig3).

RESPONSE #1-1

> Previously, we found that CD10 expression in a well-defined mesenchymal cell population beneath the nail matrix and nail bed within the nail unit, except around blood vessels and eccrine structures. Based on this finding, we proposed to name the specialized mesenchymal cells onychofibroblasts because they can be distinguished from the dermal fibroblasts by their CD10 expression. Thus, we think that CD10 is potentially expressed in other fibroblastic cell types.

Based on histomorphologic and immunohistochemical studies, we proposed the terms onychodermis (nail-specific dermis located below the nail matrix and nail bed) and onychofibroblasts (fibroblast situated within the onychodermis). The proximal portion of the onychodermis was slightly separated from the undersurface of the nail matrix by a zone of connective tissue lacking significant CD10 expression. In our human nail scRNAseq data, many CD10-positive cells were *RSPO4*-positive, however, some *RSPO4*-positive cells were CD10-negative (Fig. 2b). ISH showed *RSPO4*-positive cells in a zone of connective tissue immediately beneath the nail matrix. Considering the findings of our previous studies as well as this study, a zone of connective tissue containing *RSPO4*-positive cells right below the nail matrix should be included in the nail-specific dermis. Thus, we propose to expand the definition of the onychodermis containing onychofibroblasts to include this latter area, as delineated in Fig. 4e.

We fully agree that our study does not provide information on the two different CD10-positive cell-types other than histomorphologic and immunohistochemical information. In classifying these morphologically different CD10-positive cell-types, transcriptional profiling and functional analysis will be an important point. However, we consider it outside the scope of our current study. In the current study, we would like to focus on *RSPO4*+ onychofibroblasts and how they contribute to nail formation. Additional studies will be necessary to further explore *RSPO4*-positive cells showing two different morphology (upper spindle shaped cells versus lower round to oval shaped cells).

COMMENT #1-2

Along the same lines, most of the featureplots for onychofibroblast markers (Sfig 4) show broader expression than just cluster 4. While these could be onychofibroblastic-specific

markers, the alternate hypothesis is that these genes are heterogeneously expressed within the digit dermis. The co-labeled sections in 2f could be interpreted as heterogeneously expressed factors that are stochastically co-labeled with CD10+ cells. The colorimetric in situ make it hard to appreciate the co-localization; quantification of singly and doubly labeled cells, as well as arrows would help the reader.

RESPONSE #1-2

> We agree with some of your comment that the selected genes in the onychofibroblast cluster (supplementary figure S4) might not be the onychofibroblast-specific markers. Indeed, *TWIST1*, *CRABP1*, and *SFRP2*, which show broader expression than just cluster 4, were previously reported as the marker of skin dermis¹. Although some of these genes might not be the onychofibroblast-specific markers, it could be help in understanding characteristics of onychofibroblast. Furthermore, as we mentioned in the discussion section, *MSX1*, *WIF1*, and *RSPO4* are related to nail abnormalities, implicating that these genes might play a key role in nail formation and maintenance. Of note, cluster 4 (onychofibroblast cluster) demonstrated high expression of *RSPO4*. ISH demonstrated *RSPO4* mRNA expression restricted to mesenchymal cells below the nail matrix and nail bed with no expression in dermal fibroblasts elsewhere around the nail unit.

In the first revision, we showed the co-labeling of CD10+ cells with *RSPO4*, *BMP5*, *MSX1*, and *WIF1*. Many cells showed double labeling (brown color: CD10, red color: other markers). Following the reviewer's suggestion, arrows with co-localization information have been now included in the section describing the RNA ISH and IHC results. Quantification of singly and doubly labeled cells seems to be very difficult.

COMMENT #2

The authors state that aberrant expression of *Spink6* was found in 20 dysmorphic nails but the data is not presented. This data is key to the authors' conclusion, thus it should be presented or the statement should be removed.

RESPONSE #2

> The corresponding part of the result was removed. Twenty nail dystrophy is a disease name, but not 20 dysmorphic nails.

COMMENT #3

Insufficient methods have been provided for the newly added nail matrix keratinocyte experiments. How were fibroblasts excluded from the culture? How were the doses of recombinant proteins chosen (the concentrations of RSPO4 seem high)?

RESPONSE #3

> We appreciate the comment and have added the following in the method section.

“Briefly, after removing the lower portion from nail matrix tissue, primary nail matrix keratinocytes (NMKs) were harvested from nail matrix biopsy specimens. NMKs were isolated after 0.25% trypsin–EDTA treatment for 30 minutes and cultured in Keratinocyte Growth Media (KGM, Lonza) and maintained at 37 °C with 5% CO₂ as described previously². We checked no fibroblast growth under inverted microscope.”

We chose the doses of recombinant proteins based on the references related to the treatment of those proteins in cell culture experiments.

Reference

1. Janson, D. G., Saintigny, G., van Adrichem, A., Mahé, C. & El Ghalbzouri, A. Different gene expression patterns in human papillary and reticular fibroblasts. *J Invest Dermatol* **132**, 2565-2572 (2012).
2. Nagae, H., Nakanishi, H., Urano, Y. & Arase, S. Serial cultivation of human nail matrix cells under serum-free conditions. *J Dermatol* **22**, 560-566 (1995).

Reviewer #2 Review

I have carefully examined the revised paper. The authors have improved expression analyses on human nail tissue sections in this revision in response to my comments. I consider provided information from single cell RNA seq analyses coupled with histochemical validation is unique and valuable, particularly considering the rare availability of these human nail tissue samples. However, the manuscript still remains entirely descriptive without an experiment to functionally validate their theory. As the authors can culture human nail matrix cells and fibroblasts, there are multiple approaches with which they can at least partly validate their theory to concretely demonstrate a novel example of epidermal-fibroblast interaction. In my view, the paper would

be beneficial to a specialized scientific community in dermatopathology/dermatology.

RESPONSE:

> In the first revision, response to your comments, we performed additional experiments and made substantial changes to the text and figures to make sure the claims are well founded and clearly presented. We added functional analysis (*RSPO4* nail matrix keratinocyte analysis) showing that *RSPO4*-treated NMKs exhibited significantly higher level of *Lgr6* and *FOXQ1* than control NMKs, suggesting upregulation of WNT signaling. We also added RNA sequencing data in onychomatricoma showing similar trends of the changes of WNT signaling associated genes, *LGR6* and *FOXQ1*, which suggest mechanistic insights through epithelial-mesenchymal interactions in the nail unit. We believe this study shed light on nail biology and could help aid further functional investigations. However, as you declared, we were unable to experimentally confirm the apparent interaction between nail matrix cells and onychofibroblasts. We plan to perform co-culture experiments of human nail matrix cells and onychofibroblasts to concretely demonstrate our theory.

Reviewer #3 Review

The authors addressed most of the concerns brought up in the previous review. Previously, they identified a group of *RSPO4*+ onychofibroblasts by scRNA-seq but didn't address the functional importance of this group of cells. Now they included the observation that *RSPO4*+ cells directly underlie *Lgr6*+ NMKs (nail matrix keratinocyte) in which *B-catenin* and *Lef1* are also upregulated. They thus hypothesized that *RSPO4*+ onychofibroblasts induce Wnt signaling in *Lgr6* cells. To validate the hypothesis, they treated NMK cell culture with *RSPO4* and conducted RNA-seq. The results showed that *RSPO4*-treated NMKs exhibited significantly higher level of *Lgr6* and *FOXQ1* (Wnt marker in solid tumor) than control NMKs. The result not only suggests that *RSPO4* secreted by onychofibroblasts is important for *Lgr6* level in NMKs, but also that overexpression of *RSPO4* might be linked to onychomatricoma, a form of benign nail tumor. In general, there were improved annotations and insets that make their working model clearer to the reader. This study provides a helpful study and resource in a field that is less well-characterized than other skin appendages and can help aid further functional investigations.

COMMENT #1

In line 228 of text 'dose-dependent' effect of RSPO4 was mentioned, although it's not clear to me how figure 6d would suggest that the Lgr6 level is different between 200ng/ml and 1000 ng/ml treatment groups.

RESPONSE #1

> We thank the reviewers for the detailed comments and suggestions. The "dose-dependent" statement was toned down and removed.

COMMENT #2

Lef1 is not shown in Figure 6. Can the authors show Lef1 status since it is the Wnt marker used in Figure 4 and 5 and is a more well-established Wnt target gene.

COMMENT #3

proliferation marker/assays can be used to show that RSPO4 treatment induces cancer-like characteristics in NMKs.

COMMENT #4

The marker HEY1 is highlighted in figure 6 but not explained in text.

RESPONSE #2–4

> Following the reviewer's suggestion, the mRNA expression status of *LEF1*, *MKI67*, and *TOP2A* has now been included in Figure 6. *HEY1* does not seem to be important in our manuscript. So, we removed HEY1 data in figure 6d. The figure and legend have been updated accordingly.